# OpenReview forum: "Defeating Cerberus: Concept-Guided Privacy-Leakage Mitigation in Multimodal Language Models"
_ICLR.cc/2026/Conference — ICLR 2026 Conference Withdrawn Submission_

### Official Review · Reviewer_p7pb · 2025-10-24

**Soundness:** 2
**Presentation:** 2
**Contribution:** 2
**Rating:** 4
**Confidence:** 4

**Summary:**

This paper proposes concept-guided privacy leakage mitigation method for MLLMs, specifically focusing on VLMs. The authors identify internal concept directions relatd to PII and steer model weights along these directions to reduce privacy leakage. They also construct two multimodal PII datasets (PII-Table and CelebA-Info) and report strong refusal rates without retraining.

**Strengths:**

- The paper is clearly written and follows a standard empirical format.

- The problem of privacy-violation in MLLMs is important and socially relevant.

- The experiments are relatively comprehensive across multiple MLLMs.

**Weaknesses:**

1. The inclusion of VHTest is appreciated, as it provides a sanity check on false refusals for non-PII tasks. However, VHTest is a small-scale, purpose-built dataset rather than a general multimodal benchmark. To convincingly demonstrate that the proposed steering does not harm general utlity, I strong recommend evaluating on one more widely-recognized multimodal benchmarks.

2. The paper references activation-space steering methods such as Representation Engineering and Refusal is a Direction, which operate on hidden-state representations. However, the proposed method alters model weights instead, without any theoretical argument or empirical validation explaining why manipulating weights should preserve or even correspond to the same conceptual directions discovered in activation space. This conceptual leap—from activation-space intervention to weight-space modification—requires stronger justification. Without analysis of stability, representation drift, or equivalence between these spaces, the method feels ad-hoc rather than principled.

3. The dataset contribution is marginal. Both PII-Table and CelebA-Info are synthetic repackaging of existing text-based PII datasets converted into image form, or face-text composites built from public datasets like CelebA. There is no new data collection, annotation, or benchmarking protocol. Comparable multimodl PII datasets already exist. Thus, the dataset component mainly serves as a format conversion utility rather than a research contribution.

**Questions:**

1. Could the authors evaluate their steering method on more widely-recognised multimodal benchmarks (e.g., VQA, OCR, captioning, multi-task perception) rather than just their constructed PII-specific datasets? Including one or two general benchmarks would strengthen the claim that the mitigation preserves general model utility.

2. Beyond privacy/PII leakage, does the proposed weight-steering intervention affect other safety behaviors (e.g., refusal consistency, toxicity, jailbreak vulnerability)? Have the authors tested whether the intervention inadvertently degrades other safety mechanisms or introduces new risks?

---

### Official Review · Reviewer_QMQh · 2025-10-29

**Soundness:** 2
**Presentation:** 2
**Contribution:** 2
**Rating:** 2
**Confidence:** 4

**Summary:**

This paper introduces a method to prevent the disclosure of personally identifiable information (PII) in responses generated by a VLM, given the input image containing PII. The authors construct a multimodal privacy dataset consisting of images either directly converted to a visual format or created by superimposing a relevant image onto the textual information. The authors then use this dataset for their internal concept steering method, which computes directional steering vectors using positive and negative examples (safe and unsafe examples) based on the concept of PII. These vectors are computed layer-wise and are specifically used to steer the last few layers towards understanding unsafe disclosures. The authors then test their method on a PII-heavy VQA task and compare it to prompting baselines, finding that their method performs better than these simple baselines across model families. Finally, the authors conduct various analyses, such as understanding the effect of the coefficient applied to the steering vector and determining what types of PII are most successfully protected.

**Strengths:**

1. The authors target an important problem of protecting the leakage of PII of multimodal LLMs. The authors are correct to note that most of the work in PII protection has focused on LLMs, not VLMs.

2. The proposed method clearly and demonstrably beats the baseline prompting defenses without drastically hurting overall model performance.

3. The authors propose an interesting concept steering method based on representation engineering to guide models towards refusing to disclose PII.

**Weaknesses:**

1. The abstract claims that the method does not require fine-tuning or retraining, but weight updates are mentioned in Section 4.2. While I agree that this is not supervised fine-tuning, any weight updates should be considered training, and therefore, this claim about not requiring fine-tuning is quite misleading.

2. From weakness 1, I think that many baselines are missing, including direct SFT, DPO on the positive and negative examples, few-shot prompting, and prompting with explicit guidelines about how the PII should look. Specifically, I think it would be necessary to show that you outperform SFT and DPO and/or show that your method is less computationally expensive.

3. I looked in the appendix, but it seems like the prompt used for the prompting baselines is not provided. Could you please provide the prompt? I am worried the prompt is too generic, and so the baselines are weak.

4. In Lines 311 - 314, you mention how you only measure whether the model refuses to respond to the query. However, it seems like such a task would be agnostic of the actual input image, i.e., you could run an LLM to refuse questions without looking at the image. Could you clarify this? Do you need the image context to make abstention decisions?

5. The authors mention that sometimes models will respond with benign information, but not outright refuse, and that this is counted as a non-refusal. I think this isn't a fair analysis, and may be artificially pushing down your baseline performance.

6. The introduction mentions that "similar vulnerabilities in newer MLLMs are yet to be thoroughly investigated. However, there has been prior work on MLLM PII preservation, such as [1].

[1] Chen, Yang, Ethan Mendes, Sauvik Das, Wei Xu, and Alan Ritter. "Can language models be instructed to protect personal information?." arXiv preprint arXiv:2310.02224 (2023).

**Questions:**

1. How are refusals determined? Line 315 says "typical phrases" are used. However, there are no details about how this is done in the appendix. Is an LLM used?

2. You mention in line 377 that you chose baselines comparable to yours in setup since they do not require computing resources. However, it seems like your weight update concept steering algorithm would require computational resources. Could you clarify this?

---

### Official Review · Reviewer_cZm2 · 2025-10-30

**Soundness:** 3
**Presentation:** 2
**Contribution:** 2
**Rating:** 4
**Confidence:** 4

**Summary:**

This paper proposes a lightweight, training-free method to prevent VLMs from answering PII-related prompt by steering internal LLM representations toward refusal. Using activation engineering, the proposed method extracts a “PII sensitivity” direction from text-only data and apply weight edits to make models refuse PII-extraction requests from VLMs prompts. The approach achieves 93.3% average refusal on PII tasks with claimed minimal impact on benign performance.

**Strengths:**

- First attempt to create realistic multimodal PII datasets

- Clean application of LLM activation steering to a VLM problems

- Strong refusal results with 93.3% average refusal across PII-Table, CelebA-Info, and scanned variants, while maintain low false positives on VHTest and non-PII DocVQA (<5%).

**Weaknesses:**

- Core technique (PCA on activation diffs → weight edit) is very similar to LLM activation steering [1]. There is almost no new theoretical or algorithmic contribution. Actually the whole concept of this paper is an application of LLM activation steering to PII problem (which could be seen as a problem of alignment research)

- The utility evaluation is incomplete. While low non-refusal on non-PII datasets is reasonable to some extent, it can not accurately measure the trade-off in model utility. For example, by changing the weights, it still can answer the non-PII tasks but we are not sure about the quality of the answer. Additionally, the OCR performance shouldn't be excluded as stated in the paper. Can we test the performance on general VLM benchmarks before and after applying the steering.

- The evaluation of PII datasets is mostly generated data. While this is somewhat fine as the first investigation, the real data should be majorly considered to realistically evaluate this problem. Can we evaluate the proposed method on more real samples?


[1] Arditi, Andy, et al. "Refusal in language models is mediated by a single direction" NeurIPS 2024

**Questions:**

See weaknesses

---

### Official Review · Reviewer_Vqu9 · 2025-10-31

**Soundness:** 2
**Presentation:** 2
**Contribution:** 2
**Rating:** 2
**Confidence:** 4

**Summary:**

Summary: This paper presents a concept-guided mitigation approach that identifies and modifies the model’s internal states associated with PII-related content to mitigate the PII leakage attacks in Vision Language Models, without fine-tuning the model.

**Strengths:**

1.	This paper introduces a dataset including PII with existing dataset sources for evaluating the security measure of VLMs under the proposed defense.
2.	It proposed a mitigating technique against the PPI leakage attack by modifying the model’s internal state that triggers the model to reveal PIIs. It does not require further fine-tuning, which contributes to a lightweight solution.

**Weaknesses:**

1.	The rationale behind the lines 262-264 is not clear enough. How does the model become less inclined to reveal the PII upon modifying the model’s weight in the direction V? What does this direction vector V mean?
2.	Since the proposed method modifies the model weights, what’s the difference of fine-tuning it? What are the benefits of it instead of fine-tuning? What’s the difference in PII refusal rate for the proposed method and fine-tuning?
3.	The evaluation has been performed only on open-source small models. How would it be employed for the closed-sourced production models, such as GPT-5, Gemini-2.5?
4.	This paper mentions several jailbreaking and backdoor attacks in lines 59-60. Against which specific PII leakage attack has this defense been proposed/evaluated?
5.	How does the proposed defense impact the model’s performance/utility?

**Questions:**

Please follow the Weaknesses and respond to them.

---

### Note · Authors · 2025-11-27

I have read and agree with the venue's withdrawal policy on behalf of myself and my co-authors.